# Geographic Variations in the Toxin Profile of the Xanthid Crab *Zosimus aeneus* in a Single Reef on Ishigaki Island, Okinawa, Japan

**DOI:** 10.3390/md19120670

**Published:** 2021-11-26

**Authors:** Yuchengmin Zhang, Hideto Tsutsui, Nobuhiro Yamawaki, Yasuhiro Morii, Gregory N. Nishihara, Shiro Itoi, Osamu Arakawa, Tomohiro Takatani

**Affiliations:** 1Graduate School of Fisheries and Environmental Sciences, Nagasaki University, Bunkyo-machi 1–14, Nagasaki 852-8521, Japan; laozhangyu1993@gmail.com (Y.Z.); arakawa@nagasaki-u.ac.jp (O.A.); 2Faculty of Fisheries, Nagasaki University, Bunkyo-machi 1–14, Nagasaki 852-8521, Japan; blacksand@mail.goo.ne.jp (H.T.); yamawaki@nagasaki-u.ac.jp (N.Y.); yasu-m@nagasaki-u.ac.jp (Y.M.); 3Institute for East China Sea Research, Graduate School of Fisheries and Environmental Sciences, Organization for Marine Science and Technology, Nagasaki University, Nagasaki 851-2213, Japan; greg@nagasaki-u.ac.jp; 4Department of Marine Science and Resources, Nihon University, Fujisawa 252-0880, Japan; sitoi@nihon-u.ac.jp

**Keywords:** paralytic shellfish toxin, tetrodotoxin, xanthid crab, *Zosimus aeneus*, stomach content, morphometric observation, ascidian spicules

## Abstract

Toxic crabs of the family Xanthidae contain saxitoxins (STXs) and/or tetrodotoxin (TTX), but the toxin ratio differs depending on their habitat. In the present study, to clarify within reef variations in the toxin profile of xanthid crabs, we collected specimens of the toxic xanthid crab *Zosimus aeneus* and their sampling location within a single reef (Yoshihara reef) on Ishigaki Island, Okinawa Prefecture, Japan, in 2018 and 2019. The STXs/TTX content within the appendages and viscera or stomach contents of each specimen was determined by instrumental analyses. Our findings revealed the existence of three zones in Yoshihara reef; one in which many individuals accumulate extremely high concentrations of STXs (northwestern part of the reef; NW zone), another in which individuals generally have small amounts of TTX but little STXs (central part of the reef; CTR zone), and a third in which individuals generally exhibit intermediate characteristics (southeastern part of the reef; SE zone). Furthermore, light microscopic observations of the stomach contents of crab specimens collected from the NW and CTR zones revealed that ascidian spicules of the genus *Lissoclinum* were dominant in the NW zone, whereas those of the genus *Trididemnum* were dominant in the CTR zone. Although the toxicity of these ascidians is unknown, *Lissoclinum* ascidians are considered good candidate source organisms of STXs harbored by toxic xanthid crabs.

## 1. Introduction

Toxic crabs of the family Xanthidae accumulate saxitoxins (STXs) and/or tetrodotoxin (TTX) and pose a threat to human health [1] (Figure 1). STXs comprise a group of neurotoxins produced by toxic dinoflagellates and cyanobacteria and are involved in the toxification of bivalves [2]. The molecular weight, toxicity, and intoxication mechanism of the best-known component of STXs, STX, are similar to those of the pufferfish toxin, TTX [3]. Poisoning incidents due to toxic xanthid crabs are reported around the world, including Japan, Australia, Taiwan, and the Philippines, and several epidemiologic investigations, including identification of the causative toxin, have been conducted [4,5,6,7,8,9,10,11].

The toxicity and toxin profile of xanthid crabs vary greatly depending on the species and/or habitat. *Zosimus aeneus*, *Atergatis floridus*, and *Platypodia granulosa*, which inhabit coral reefs in the Ryukyu and Amami Islands in Japan, the Philippines, and Australia, are generally highly toxic and contain STXs as their main toxin [4,7,11,12], whereas *Z. aeneus* from the Tokara Islands in Japan; *A. floridus* from the Pacific coast of mainland Japan; *Z. aeneus*, *A. floridus*, *Lophozozymus pictor*, and *Demania reynaudi* from Taiwan; *Demania cultripes* from Cebu Island in the Philippines, are relatively less toxic and have TTX as their major toxin component [9,10,13,14,15]. In Australia, the toxicity of *A. floridus* varies significantly depending on the collection area and season [7]. In Ishigaki Island, Okinawa Prefecture, Japan, the toxin composition of *A. floridus* is reported to differ among two reefs separated by a passage and between a reef and the small islet Kojima, which is adjacent to the reef [11,13]. The toxin profile of xanthid crabs, however, also varies greatly among individuals, making it difficult to elucidate the details of small-scale within reef variation.

Pufferfish, which, like xanthid crabs, contain both STXs and TTX, are considered to ingest and accumulate these toxins from toxic prey such as starfish, shellfish, and flatworms [16,17]. Xanthid crabs, even within the same species, are assumed to be exogenously toxified through the food chain, like pufferfish, because there are great individual and regional variations in the toxin profile of these crabs. Kotaki et al. [18] and Yasumoto et al. [19] reported the detection of STXs and TTX in the red alga *Jania sp.* distributed in the habitat of toxic xanthid crabs and speculated it as the origin of the crab toxin. It is unlikely that this red alga alone can account for the extremely high STXs content of xanthid crabs in the Ryukyu Islands. Therefore, some unknown toxic organisms that accumulate high concentrations of STXs are hypothesized to be present in the coral reefs inhabited by toxic xanthid crabs.

In our study, to clarify small-scale within reef variation in the toxin profile of xanthid crabs, we collected a total of 53 *Z. aeneus* specimens on two occasions in 2018 and 2019 within a Yoshihara reef off the coast of Yoshihara, Ishigaki Island, and recorded the GPS coordinates of each individual (Figure 2). The STXs/TTX content of the appendages and viscera or stomach contents were quantified then visualized to show the characteristic toxin profiles in specific sampling zones. In addition, the stomach contents of crab specimens collected in zones with different toxin profiles were examined and compared to estimate the source organisms of the crab toxins, especially STXs.

## 2. Results

### 2.1. Toxin Profile of Z. aeneus Collected in 2018

STXs (5.84–7290 nmol/g) were detected in the appendages of 19 of the 28 tested individuals, and TTX (0.29–75.9 nmol/g) was detected in 24 of the 28 tested individuals (Figure 3A). Neither STXs nor TTX was detected in one non-toxic individual (No. 24). Of the 27 toxic individuals, approximately 60% (16 individuals; No. 1–4, 7–10,15,16,20–23, 25, and 26) contained both STXs and TTX, while three individuals (No. 5, 6, and 27) contained only STXs, and eight individuals (No. 11–14, 17–19, and 28) contained only TTX. Given the ratio of STXs/TTX, STXs accounted for more than 94% of the total toxin content in toxic individuals, except for the eight individuals containing only TTX and No. 9, which contained a relatively high TTX content (64% STXs). STXs include five components of the STX group (neoSTX, hyneoSTX, STX, hySTX, and dcSTX), and at least four components of the gonyautoxin (GTX) group (GTX2, GTX3, dcGTX2, and dcGTX3). The ratio of neoSTX+hyneoSTX and STX+hySTX+dcSTX varied among individuals (10–90% neoSTX+hyneoSTX and 2–97% STX+hySTX+dcSTX), but the STX group accounted for more than 90% of the total toxin content (STXs+TTX) in many individuals. In xanthid crabs collected in 2018, STXs (40.8–7750 nmol/g) were detected in the viscera of 15 of the 28 tested specimens, and TTX (0.32–35.2 nmol/g) was detected in 24 of the 28 tested specimens (Figure 3B). The toxin content of the viscera was generally similar to that of the appendages. Among the 19 individuals with STXs in the appendages, four (No. 3, 15, 20, and 21) showed no STXs in the viscera. A total of 12 individuals contained both STXs and TTX or only TTX in the viscera. In the individual with non-toxic appendages (No. 24), the viscus was also non-toxic. The ratio of STXs/TTX in the viscera was generally similar to that in the appendages, except for the four individuals in which STXs were detected in the appendages but not in the viscera. STXs comprising mainly the STX group accounted for more than 86% of the total toxin content, except for 12 individuals containing only TTX and No. 10, which had a relatively high proportion of the GTX group in addition to TTX.

The toxin profile (pattern of STXs/TTX content and composition) shown in Figure 3 appears to differ among the northwestern, central, and southeastern parts of the reef (designated NW, CTR, and SE zones, respectively) (Figure 2). For example, in the NW zone, there were many individuals with a very high content of STXs in both the appendages and viscera; the appendages of six and viscera of three of eight individuals had STXs content exceeding 1000 nmol/g, whereas the highest STXs content detected in the CTR zone was only 252 nmol/g. STXs were not detected in the appendages of seven or in the viscera of eight of eleven individuals. However, in the CTR zone, TTX was detected in both the appendages and viscera of all individuals, and the TTX content seemed to be slightly higher than that in the other zones. Regarding the toxin composition in general, the NW and SE zones were dominated by STXs, while the CTR zone was dominated by TTX. Statistical analysis revealed a significant difference in the STXs content of the appendages and viscera of crabs collected from the three zones (Kruskal–Wallis test, *p* < 0.01), and although no significant difference was detected between the SE and NW or CTR zones, the STXs content of specimens from the NW zone was significantly higher than that of specimens from the CTR zone (Dunn’s post hoc test, *p* < 0.01). On the other hand, there was no significant difference in the TTX content of the appendages among crabs from the three zones, but the TTX content in the viscera was significantly higher in specimens from the CTR zone than in specimens obtained from the other two zones (*p* < 0.05).

### 2.2. Toxin Profile of Z. aeneus Collected in 2019

The toxin content and toxin composition in the appendages and stomach content of each crab specimen collected in 2019 are shown in Figure 4. 

In xanthid crabs collected in 2019, STXs (5.17–3200 nmol/g) were detected in the appendages of 19 of the 25 test animals, and TTX (0.17–88.5 nmol/g) was detected in the appendages of 14 of the 25 tested individuals (Figure 4A). There was one non-toxic individual (No. 38). Of the 24 toxic individuals, nine individuals (No. 29, 35, 36, 39–41, 45, 48, and 53) contained both STXs and TTX (a lower number of crabs containing both toxins in the appendages compared with those collected in 2018), while ten individuals (No. 30–34, 37, 44, 46, 51, and 52) contained only STXs, and five individuals (No. 42, 43, 47, 49, and 50) contained only TTX. With regards to the toxin composition, STXs accounted for more than 92% of the total toxin content in the toxic individuals, except for No. 40 and 41 (72% and 46% STXs, respectively), and the five individuals containing only TTX. The STXs components were the same as those in specimens collected in 2018, with the STX group being predominant in many individuals, but in some individuals from the SE zone, the GTX group was predominant.

STXs (2.89–1247 nmol/g) were detected in the stomach contents of 17 of the 25 tested animals, and TTX (0.06–2.76 nmol/g) was detected in 12 of the 25 tested specimens (Figure 4B). The amount of toxin in the stomach contents was generally lower than that in the appendages. Of the 19 individuals in which STXs were detected in the appendages, three (No. 39–41) exhibited no STXs in the stomach contents, and of the five individuals in which only TTX was detected in the appendages, one (No. 50) showed a trace amount of STXs in the stomach contents. On the other hand, of the 14 individuals in which TTX was detected in the appendages, five (No. 39, 41, 43, 45, and 50) exhibited no TTX in the stomach contents, and of the ten individuals in which only STXs were detected in the appendages, two (No. 34 and 46) exhibited only small amounts of TTX in the stomach contents; one individual (No. 38) whose appendages were non-toxic showed a small amount of TTX in the stomach contents. Therefore, seven individuals (No. 29, 34–36, 46, 48, and 53) showed both STXs and TTX in the stomach contents, ten individuals (No. 30–33, 37, 44, 45, and 50–52) showed only STXs, and five individuals (No. 38, 40, 42, 47, and 49) showed only TTX. The ratio of STXs/TTX in the stomach contents was similar to that in the appendages, except for in No. 40, in which STXs were detected only in the appendages, and No. 50, in which STXs were detected only in the stomach contents. The composition of STXs in the stomach contents, however, differed from that in the appendages; the GTX group was predominant in many individuals.

The characteristics of the toxin profiles for each zone were similar to those of specimens collected in 2018; many individuals from the NW zone had a very high amount of STXs in both the appendages and stomach contents, whereas in the CTR zone, only small amounts of STXs were detected in some individuals, and in the SE zone, the characteristics were intermediate between the other two zones. Statistical analysis revealed that the STXs content of the appendages was significantly higher in specimens from the NW and SE zones than in specimens collected from the CTR zone (*p* < 0.01 and *p* < 0.05, respectively) and that the amount of STXs in the stomach contents was significantly higher in specimens from the NW zone than in specimens collected from the CTR zone (*p* < 0.01). The amount of TTX in the appendages and stomach contents did not differ significantly among the three zones (*p* > 0.05).

### 2.3. Analysis of Stomach Contents

In 2019, three *Z. aeneus* specimens were collected from each of the NW and CTR zones, separately from those specimens used for the toxin profile analysis (Figure 2), and the stomach contents were analyzed. Under light microscope observation, we found large numbers (a total of 1263 and 1676 counts, respectively) of ascidian spicules from both zones, in addition to small numbers of foraminifers, sponge spicules, algae, and mollusk fragments (Figure 5). The ascidian spicules found were classified into four types (As-1 to As-4) according to their morphology and previous reports [20,21,22]. Type As-1 to As-3 were identified as the genus *Trididemnum*, and type As-4 was identified as the genus *Lissoclinum*. In the stomach contents of the crab specimens collected from the NW zone, ascidian spicules of type As-4 were predominant, whereas ascidian spicules of As-1 type were most frequently observed in the stomach contents of crabs collected from the CTR zone. 

The contribution of ascidian spicules by type in each zone is shown in Figure 6A. The As-4 type was dominant in the NW zone, accounting for 92% of the ascidian spicules, whereas the As-1 type was most abundant (70%) in the CTR zone, followed by As-2 (15%) and As-3 (7%). However, 8.47 nmol/g of the GTX group was detected in the stomach contents of specimens from the NW zone, while 0.67 nmol/g of TTX was detected in the stomach contents of specimens from the CTR zone (Figure 6B).

## 3. Discussion

The present study revealed small-scale within reef variation in the toxin profile of *Z. aeneus*. We demonstrate that a single reef may contain zones that can be delineated by the toxin profiles of the inhabitants: a zone in which the crabs accumulate extremely high concentrations of STXs (NW; STXs-rich zone) and a zone in which the crabs generally contain small amounts of TTX and little STXs (CTR; STXs-poor zone). In addition, crabs living in the two zones differed in the composition of their stomach contents; ascidian spicules of the genus *Lissoclinum* dominated in the STXs-rich zone, and ascidian spicules of the genus *Trididemnum* dominated in the STXs-poor zone. Although the toxicity of these ascidians is unknown, ascidians of the genus *Lissoclinum* are considered a good candidate source of STXs harbored by xanthid crabs.

Various toxicity studies in xanthid crabs, especially *Z. aeneus* and *A. floridus*, living on the reefs around the mouth of Kabira Bay, Ishigaki Island, including Yoshihara reef, have revealed that despite the varying toxicities among individual crabs, they are generally highly toxic, with STXs (mainly STX group) as the major toxin components [4,5,6,11,12,13]. The toxin profile of crab specimens from the NW zone in this study was generally consistent with the previously reported toxin profile of crabs on these reefs. On the other hand, *A. floridus*, which inhabits a small islet (Kojima) separated from these reefs by a lagoon, is relatively less toxic and contains TTX as their main toxin [23]. The toxin profile of the crab specimens from the CTR zone is similar to that of *A. floridus* from Kojima. Previous studies examined the toxin profile of pooled samples by region rather than by individual, and therefore small-scale within reef variation may have been overlooked. It is relevant to note that *Z. aeneus* individuals from the Philippines that were STXs-dominant toxin or TTX-dominant toxin were mixed [24]; however, whether this is due to individual or spatial variation remains unclear.

In crab specimens collected in both 2018 and 2019, there was only one individual each year in which no toxin was detected in the appendages. In the toxic individuals, individuals with both STXs and TTX and individuals with either STXs or TTX coexisted, and their occurrence patterns and toxin profiles were generally consistent between the appendages and viscera in 2018 but somewhat different between the appendages and stomach contents in 2019. This may be because the toxin profile of the stomach contents is a snapshot of the toxin profile of the prey ingested by the crab at the time of sampling, whereas the toxin profile of the appendages and viscera which includes not only the stomach but also the heart, gonads, and intestine, reflects the integrated value of the toxins ingested by the crab over a long period. It is unclear how long it takes for the toxin profile of *Z. aeneus* to reflect a change in environment, given that the movement ecology of *Z. aeneus* remains to be determined. Nevertheless, the similarity in the small-scale within reef variation in the toxin profile between the 2018 and 2019 specimens suggests that the NW, CTR, and SE zones have an STXs-rich, STXs-poor, and intermediate environment, respectively.

In this study, a total of nine components, neoSTX, hyneoSTX, STX, hySTX, dcSTX, GTX2, GTX3, dcGTX2, and dcGTX3, were detected in crab specimens containing STXs (Figure 7). Of these, hyneoSTX and hySTX were isolated and identified from *Z. aeneus* collected from the reefs of Ishigaki Island and are specific to xanthid crabs (i.e., not found in other organisms) [25]. The ratio of these STXs components varied among individuals, but, as in previous reports [12], the STX group was predominant in the appendages and viscera of the majority of individuals. When *A. floridus* was administered GTX group toxins, they were converted to STX group toxins within a short period of time [26]. Two species of bacteria, *Pseudomonas* sp. and *Vibrio* sp., isolated from the viscera of *A. floridus*, are known to convert GTX group toxins to STX group toxins [27]. Therefore, we deduce that xanthid crabs ingest GTX group dominated STXs from toxic prey and convert them to STX group toxins for accumulation in their bodies. These marine bacteria have been known to have the ability to transform hydroxysulfate carbamate derivatives to STX through reductive eliminations, but little is known about the effects of toxin transformation on the crabs themselves. 

In a previous study, 11-oxoTTX and 11-*nor*TTX-6(*R*)-ol, in addition to TTX, were isolated and identified from *A. floridus* from Kojima, Ishigaki Island [28]. In the present study, only the main component, TTX, was analyzed because we have no appropriate standards for TTX derivatives. In addition to TTX, however, TTX derivatives such as 11-oxoTTX, deoxyTTXs, 4,9-anhydroTTX, and 4-*epi*TTX were also detected qualitatively at *m/z* 336 > 162, *m/z* 304 > 162, *m/z* 302 > 162, and *m/z* 320 > 162 (Figure 8). It is considered that TTX-containing organisms, such as pufferfish, accumulate TTX through the food chain, which starts with TTX-producing bacteria [29]. Although TTX-producing bacteria have been isolated from the intestine of *A. floridus* [30,31], the origin of TTX harbored by xanthid crabs, and the involvement of bacteria in the toxin accumulation, remain unclear and further studies are required.

Saisho et al. reported that the stomach contents of *Z. aeneus* from Ishigaki Island consisted of green algae, brown algae, red algae, porifera, corals, bivalves, gastropods, and sand [32]. Although macroalgae were observed, the originating organism of the crab toxin could not be identified. In strong contrast to their results, we found an abundant number of ascidian spicules, whereas the presence of other organisms was minimal. Ascidians have several forms of spicules in their bodies, and their geographic variation in morphology makes it difficult to clearly identify ascidian species from their spicules. However, according to previous reports [20,21,22,33,34] the four types of ascidian spicules found in the stomach contents of *Z. aeneus* could be classified into two genera (*Lissoclinum* and *Trididemnum*). It is relevant to note that toxic ascidians have been reported; STXs were found in *Microcosmus vulgaris* and *Holocynthia roretzi*, and STXs and TTX in *Phallusia nigra* [35,36,37]. The toxicity of the genera *Lissoclinum* and *Trididemnum* are unknown, but *Lissoclinum* ascidian spicules (As-4) were predominant, and GTX group toxins were detected in the stomach contents of the crabs collected from the STXs-rich NW zone, whereas only a few *Lissoclinum* ascidian spicules were found and no STXs were detected (possibly below the detection limit) in the stomach contents of the crabs collected from the STXs-poor CTR zone, suggesting that *Lissoclinum* ascidians are a source of the STXs. Although we were not able to collect live ascidian specimens in our study, we plan to further investigate the geographic distribution, toxicity, and toxin profile of these ascidians on the reefs of Ishigaki Island.

## 4. Materials and Methods

### 4.1. Crab Specimens

In June 2018 and June 2019, 28 and 31 individuals of *Z. aeneus* were collected, respectively, within a single reef (Yoshihara reef) off the Yoshihara coast of Ishigaki Island, Okinawa, Japan. The coordinates of the sampling location for each sample were recorded with a GPS receiver (GPSMAP® 62SJ, Garmin, Lenexa, KS, USA) (Figure 1). The 28 individuals collected in 2018 and 25 of 31 individuals collected in 2019 were transported as specimens for the toxin profile analysis to the training ship Nagasaki-Maru, Nagasaki University, moored at Ishigaki Port and immediately frozen in a freezer on board. The specimens were then transported frozen to the laboratory of the Graduate School of Fisheries and Environmental Sciences, Nagasaki University, where they were kept frozen below −20 °C until extraction and quantification of toxins.

For the remaining 6 of 31 individuals collected in 2019 (3 individuals each collected in the NW and CTR zones for stomach content analysis), the stomach was immediately separated from the body after anesthetizing with ice water for 15 min, and the contents were collected in the Nagasaki-Maru wet laboratory. The stomach contents of the 3 individuals collected in the same zone were pooled; a portion was used for stomach content analysis by light microscopy, and the remainder was used for extraction and quantification of toxins.

### 4.2. Toxin Extraction

The specimens collected in 2018 were thawed and divided into appendages and viscera (including stomach, heart, gonads, and intestine), while the specimens collected in 2019 for toxin profile analysis were divided into appendages and stomach contents. Toxins from each of these samples, as well as from the stomach contents of the specimens used for stomach content analysis, were extracted by heating with twice the amount of 0.1 M HCl for 5 min, centrifuged at 3000 × *g* for 10 min, and then the supernatant was passed through an HLC-DISK membrane filter (0.45 µm, Kanto Chemical Co., Ltd., Tokyo, Japan). The obtained test solution was subjected to toxin quantification as described below.

### 4.3. Toxin Quantification

#### 4.3.1. High-performance Liquid Chromatography with Fluorometric Detection (HPLC-FLD) for STXs

HPLC-FLD was carried out on a Waters system (Milford, MA, USA) fitted with a fluorescence detector [29,38]. A LiChroCART Superspher RP-18 (e) column (4.6 × 250 mm, Merck, Darmstadt, Germany) was used. The mobile phase for the GTX group was 2 mM heptanesulfonic acid in 10 mM ammonium phosphate buffer (pH 7.3). The mobile phase for the STX group was 2 mM heptanesulfonic acid in 4% acetonitrile-30 mM ammonium phosphate buffer (pH 7.3). All the flow rates were set at 0.8 mL/min, and the temperature of the column was set at 35 °C. The eluate from the column was continuously mixed with 0.2 M KOH containing 1 M ammonium formate and 50% formamide, and with 50 mM periodic acid at a flow rate of 0.4 mL/min each, and then heated at 65 °C for 1.5 min. The fluorophores formed were detected at excitation and emission wavelengths of 336 and 392 nm, respectively. The toxin standards used for quantification in this study were GTX1–4 and dcGTX2, 3 provided by the Japan Fisheries Research, and STX, neoSTX, dcSTX, hySTX, and hyneoSTX purified by us from the toxic crab *Z. aeneus*. The limit of detection (LOD) of STXs was 0.01–0.03 nmol/g tissue (S/N = 3), and the limit of quantification (LOQ) of STXs was 0.03–0.1 nmol/g tissue (S/N = 10).

#### 4.3.2. Liquid Chromatography-tandem Mass Spectrometry (LC-MS/MS) for TTX

LC-MS/MS was conducted according to a previously reported method [39]. LC was performed on an Alliance 2690 Separations Module (Waters). A Mightysil RP-18 GP column (2.0 × 250 mm, particle size 5 µm, Kanto Chemical Co., Inc.) was used with the mobile phase of 30 mM heptafluorobutyric acid in 1 mM ammonium acetate buffer (pH 5.0). The flow rate was set at 0.2 mL/min. The eluate was introduced into a Quattro micro^TM^ API detector (Waters). TTX was ionized by positive-mode electrospray ionization with a desolvation temperature of 400 °C, source block temperature of 150 °C, and cone voltage of 50 V, and monitored at m/z 162 (quantitative) and m/z 302 (qualitative) as product ions (collision voltage 38 V) with m/z 320 as a precursor ion through a MassLynx^TM^ NT operating system (Waters). TTX standard (FUJIFILM Wako Pure Chemical Corporation, Osaka, Japan) was used as an external standard. The LOD of TTX was 0.009 nmol/g tissue (S/N = 3) and the LOQ of TTX was 0.03 nmol/g tissue; (S/N = 10).

### 4.4. Stomach Content Analysis

The organisms obtained from the stomach contents were pretreated as follows to facilitate observation under a microscope. Approximately 0.1 g of stomach contents was placed in a 15 mL centrifuge tube, and an equal volume of kitchen drain cleaning fluid (e.g., Pipeman® Johnson and Johnson, New Brunswick, NJ, USA) was added to the sample along with the 3 volumes of distilled water. The Pipeman cleaning fluid contains a total concentration of 0.8% sodium hypochlorite, as well as sodium hydroxide, alkyl amine oxide, potassium salts of fatty acids, and 2-phenoxyethanol [40,41]. After the cleaning process, the rinsed contents were placed on a microscope cover slide (Matsunami Glass Ind., Ltd., Osaka, Japan; No. 1, 22 mm x 40 mm) and dried at room temperature. After drying, the samples were immersed in immersion oil (Olympus® IMMOIL-F30CC). A light microscope (Olympus® BH2) with a polarizer was used for observations. More than 1000 micro-materials (i.e., microscopic inorganic and organic material) were counted for each zone to capture the wide variation in material.

### 4.5. Statistical Analysis

Statistical analysis was performed using R version 4.1.0 (R Core Team 2021). A Kruskal–Wallis test was used to compare the STXs/TTX content in the appendages, viscera, or stomach contents among each of the three zones, and Dunn’s post hoc test was conducted when significant differences were detected.

## Figures and Tables

**Figure 1 marinedrugs-19-00670-f001:**
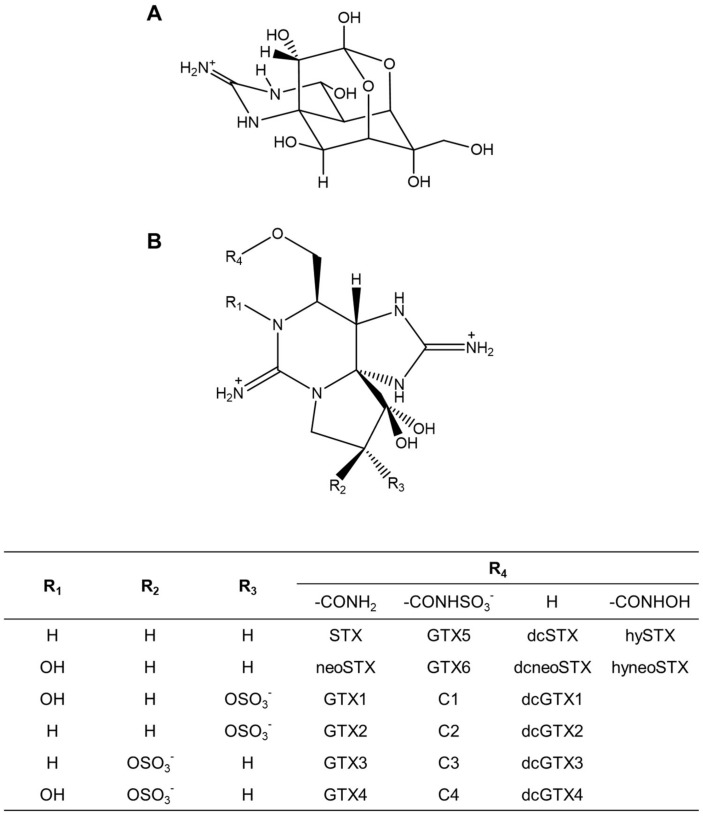
Structures of (**A**) TTX and (**B**) STXs analogues. GTX: gonyautoxin; dc: decarbamoyl; hy: carbamoyl-N-hydroxy; hyneo: carbamoyl-N-hydroxyneo.

**Figure 2 marinedrugs-19-00670-f002:**
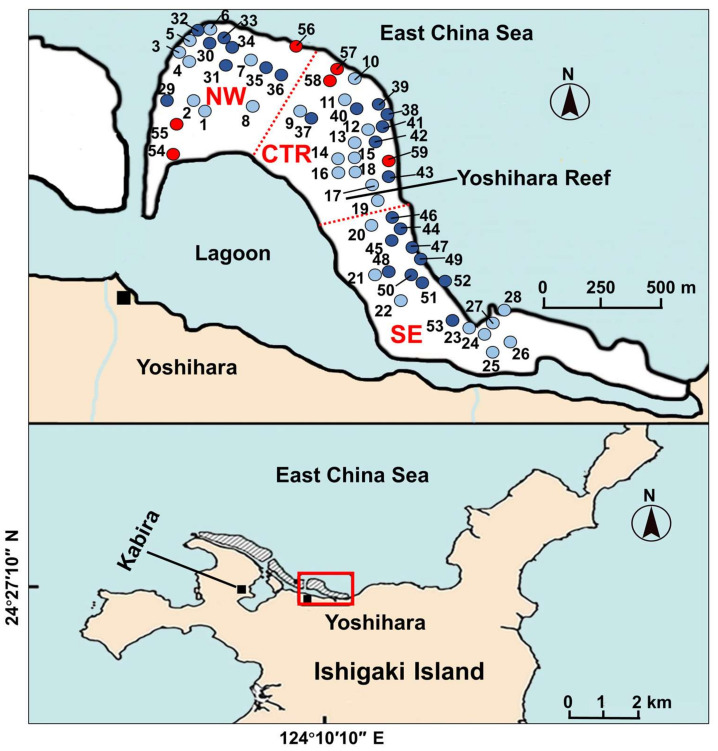
Map showing the coral reef off the Yoshihara coast (Yoshihara reef), which can be divided into northwestern, central, and southeastern parts (designated NW, CTR, and SE zones, respectively) according to the toxin profile of inhabiting crab specimens. The locations where the crab specimens were collected are indicated by circles with the specimen number. Light blue, dark blue, and red circles represent specimens collected in 2018 (No. 1–28), specimens collected in 2019 used for toxin profile analysis (No. 29–53), and specimens collected in 2019 used for the stomach contents analysis (No. 54–59), respectively.

**Figure 3 marinedrugs-19-00670-f003:**
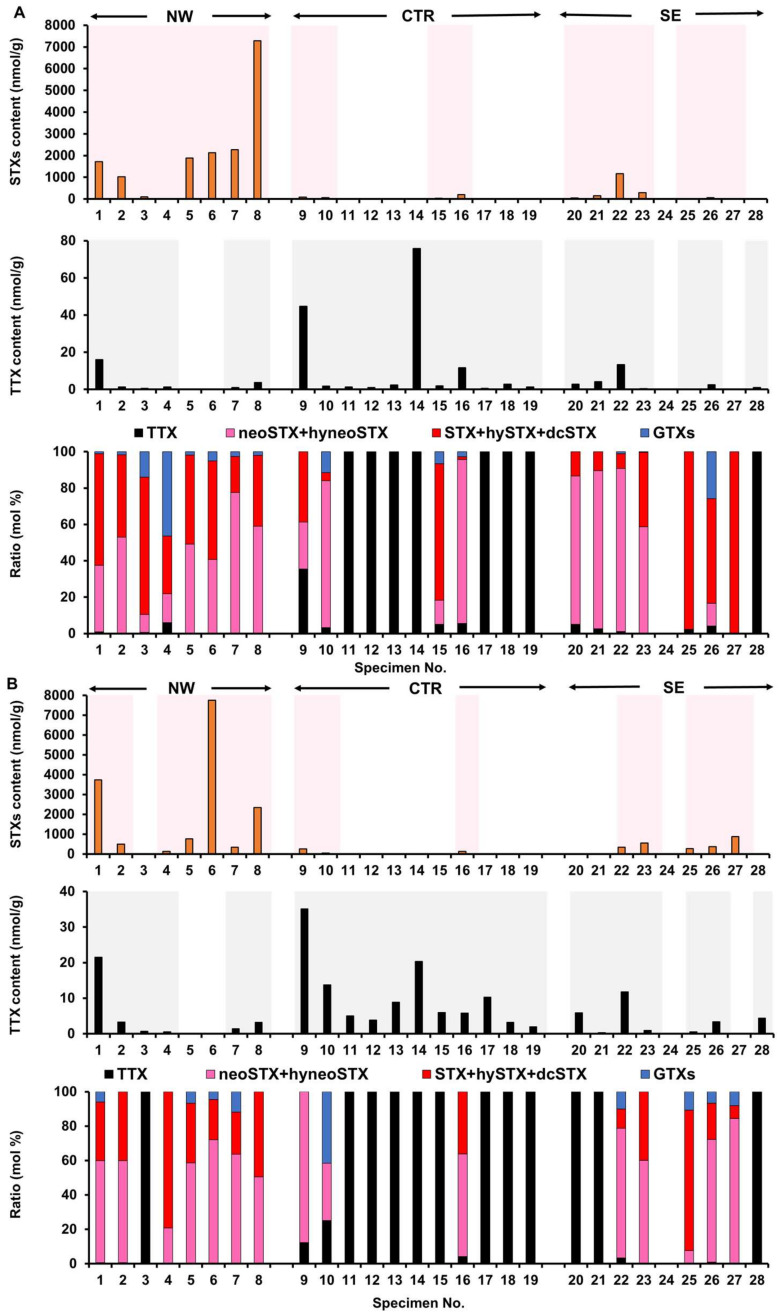
STXs and TTX content and ratio of each toxin component in the appendages (**A**) and viscera (**B**) of individual crab specimens collected in the NW, CTR, and SE zones in 2018. In the graphs representing the STXs and TTX content, the light background colors indicate toxic individuals and bars indicate the toxin content. For some toxic individuals, the toxin content was smaller than the resolution of the y-axis scale.

**Figure 4 marinedrugs-19-00670-f004:**
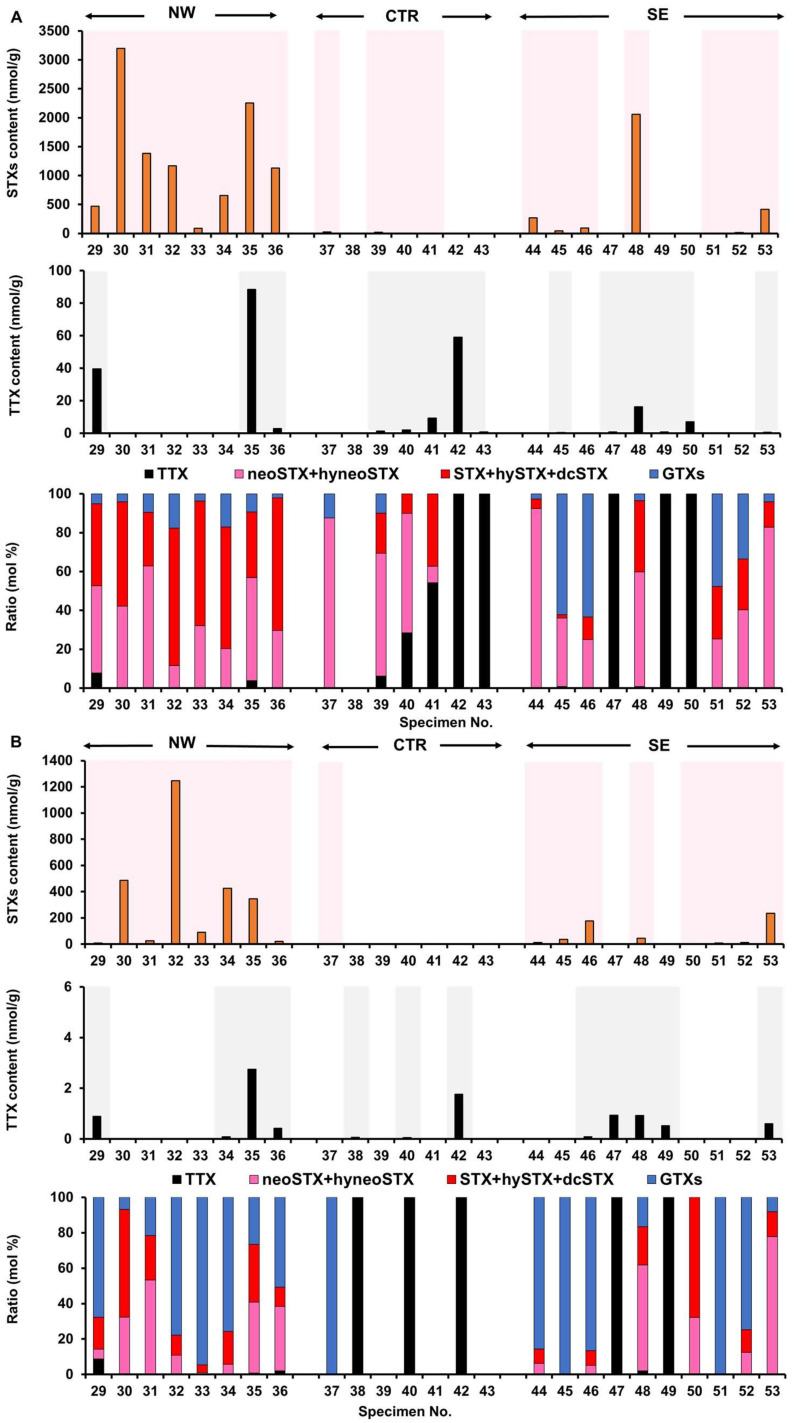
STXs and TTX content and ratio of each toxin component in the appendages (**A**) and stomach contents (**B**) of each crab specimen collected in NW, CTR, and SE zones in 2019. In the graphs representing the STXs and/TTX content, the light background colors represent individuals in which the toxin was detected, and bars indicate the toxin content. For some toxic individuals, the toxin content was smaller than the resolution of the y-axis scale.

**Figure 5 marinedrugs-19-00670-f005:**
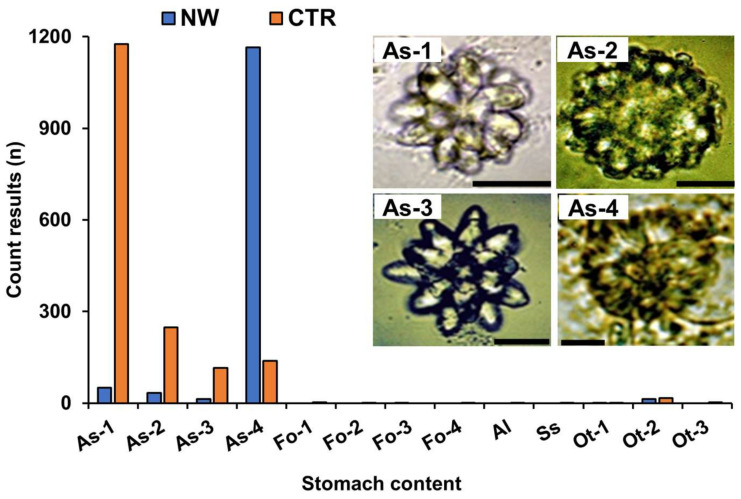
Total counts of micro-materials in the stomach contents of the crab specimens collected in NW and CTR zones. As: ascidian spicules; Fo: planktonic/benthic foraminifers; Al: algae; Ss: sponge spicule; Ot: other spicules. The photos show four types of ascidian spicules (As-1 to As-4) observed in the stomach contents. As-1, As-2, and As-3 were assigned to the genus *Trididemnum*, while As-4 was assigned to the genus *Lissoclinum*. Scale bar in the photo of As-1 indicates 20 µm; As-2 and As-3 10 µm, and As-4 5 µm.

**Figure 6 marinedrugs-19-00670-f006:**
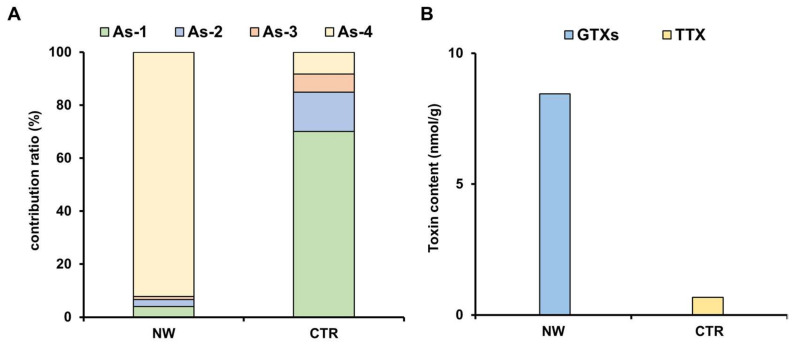
Contribution ratio of ascidian spicules by type (As-1 to As-4) observed in the stomach contents (**A**) and the toxin contents of the stomach contents (**B**) from crab specimens collected from the NW and CTR zones.

**Figure 7 marinedrugs-19-00670-f007:**
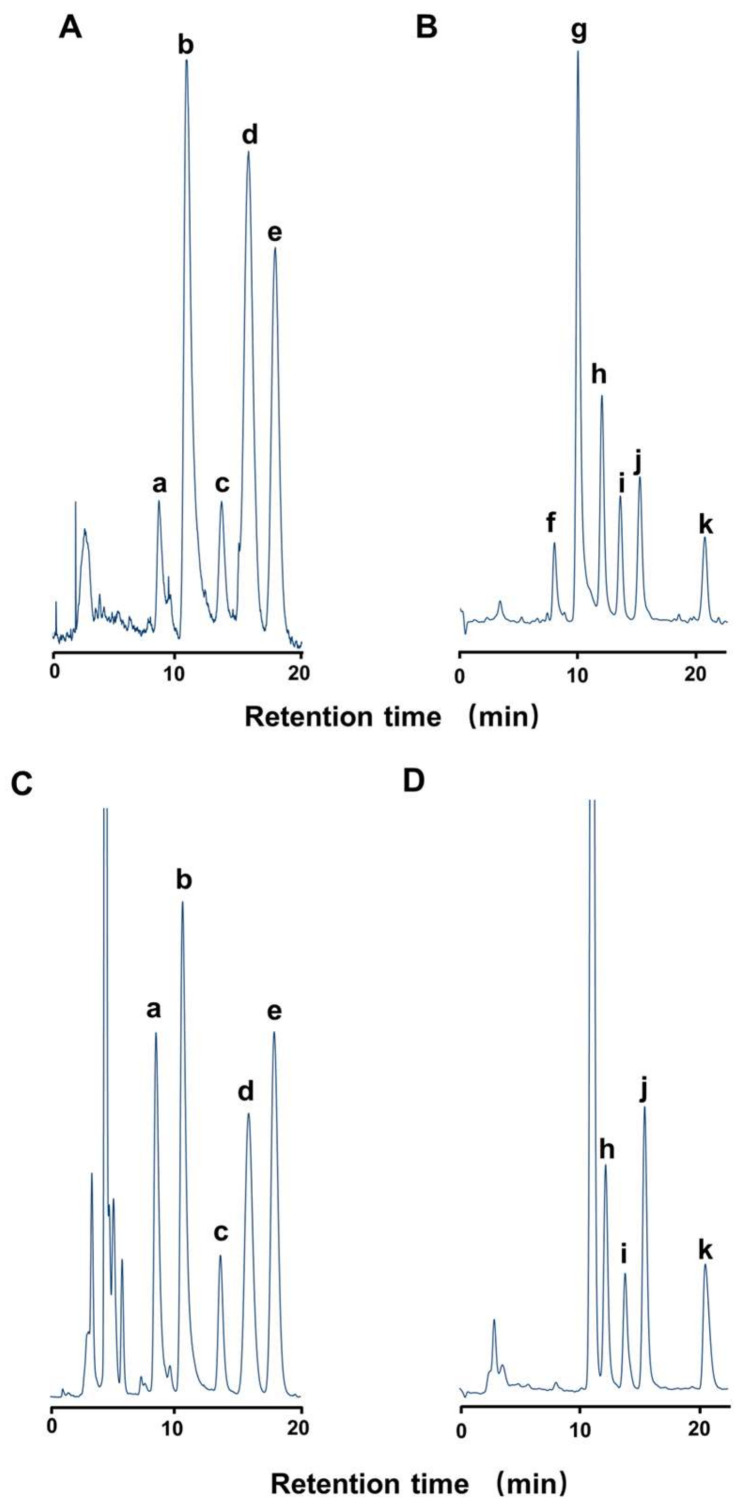
HPLC-FLD chromatograms of the toxin standards (**A**, **B**) and appendages of crab No. 48 (**C**, **D**), a: hyneoSTX; b: neoSTX; c: hySTX; d: dcSTX; e: STX; f: GTX4; g: GTX1; h: dcGTX3; i: dcGTX2; j: GTX3; k: GTX2. (**A**, **C**: STX group, **B**, **D**: GTX group).

**Figure 8 marinedrugs-19-00670-f008:**
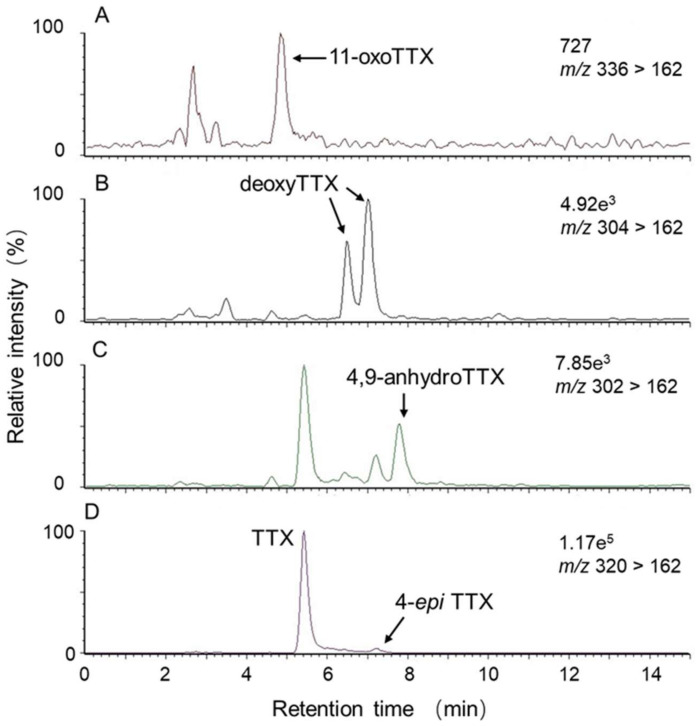
LC-MS/MS chromatograms at *m*/*z* 336 > 162 (**A**), *m*/*z* 304 > 162 (**B**), *m*/*z* 302 > 162 (**C**), and *m*/*z* 320 > 162 (**D**) of the appendages extract from specimen No. 35. TTX: 85%; 11-oxoTTX: 1%; deoxyTTX: 6%; 4,9-anhydroTTX: 4%; 4-*epi*TTX: 4%. The percentages indicated the content of the TTX and TTX-related substances, assuming that their ionization efficiency per unit amount was the same as that of TTX.

## Data Availability

The data that support the findings of this study are available from the corresponding author upon reasonable request.

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
