# Peer review of "Geographic Variations in the Toxin Profile of the Xanthid Crab Zosimus aeneus in a Single Reef on Ishigaki Island, Okinawa, Japan"

_marinedrugs, 2021, doi:10.3390/md19120670_

Round 1
Reviewer 1 Report
The manuscript reported geographic variations in the STXs/TTX profile of the xanthid crab Zosimus aeneus in a single reef. Although the presence of ascidian spicules was observed in the stomach contents of crab specimens, this cannot be inferred that the toxin originated from the ascidian. Toxin profiles in ascidian were not analyzed in this manuscript, and toxins in crabs may originate from toxin-producing microalgae or other organisms. Why were the toxin profiles in the crabs collected from the three regions obvious different? STX and TTX are two different groups of toxins, the ratio of STXs/TTX in sample has no senses. In the toxin analysis method, parameters for method evaluation are lacking, such as matrix effects, recovery. In 4.3.1, which STX components were analyzed in HPLC-FLD method.
Author Response
To Reviewer 1: Thank you very much for your valuable comments. We would like to respond to your comments below.
The manuscript reported geographic variations in the STXs/TTX profile of the xanthid crab Zosimus aeneus in a single reef. Although the presence of ascidian spicules was observed in the stomach contents of crab specimens, this cannot be inferred that the toxin originated from the ascidian. Toxin profiles in ascidian were not analyzed in this manuscript, and toxins in crabs may originate from toxin-producing microalgae or other organisms. Why were the toxin profiles in the crabs collected from the three regions obvious different?
>>Reply: Thank you for your valuable comments. In 2018, we analyzed the toxin profile and toxin content in the xanthid crab Zosimus aeneus and the distinct individual and spatial variations in a single reef were found. The result suggested the crabs were exogenously toxified through the food web. Therefore, we analyzed the stomach content instead of viscera in 2019 and tried to elucidate the diet of Zosimus aeneus. Toxins were detected in the stomach contents and an abundant number of ascidian spicules were found. As indicated in the text, the composition of the stomach contents of the crab differed greatly between the two zones, as did their toxin components. Since the stomach contents of crabs were of a small amount, it was not possible to analyze the individual ascidian toxins contained in the stomach contents. The red alga Jania sp. has been reported as one of the origins of crab toxins, but since Jania was not found in the stomach contents in our survey, we suspect the existence of other toxic prey organisms without Jania, and we believe that ascidians are one of the candidates. (This is not a definite conclusion.) We have not been able to acquire any ascidians of the same species from the field, so we plan to investigate further.
STX and TTX are two different groups of toxins, the ratio of STXs/TTX in sample has no senses.
>>Reply: The main toxin components of xanthid crabs vary from region to region, and some individuals have been reported to have both STXs and TTX. The molecular weight, toxicity, and intoxication mechanism of STX and TTX are similar, pufferfish are known to possess proteins (PSTBP) that specifically bind both toxin components. In addition, just as it was found, for example, that there is a difference in the selectivity of STXs and TTX between marine and freshwater pufferfish, we believe that similar studies and discussions will be necessary for toxic crabs in the future. For this reason, we believe it is necessary to compare both STXs and TTXs in this paper.
In the toxin analysis method, parameters for method evaluation are lacking, such as matrix effects, recovery.
>>Reply: As a way to prevent matrix effects, we diluted the samples when analyzing them. Due to the scarcity of toxin standards, we have not performed any operation to determine the recovery rate. However, the LC-MS/MS and HPLC-FLD methods used in this study have been used in the analysis of many TTX and STXs, and are highly reliable.
In 4.3.1, which STX components were analyzed in HPLC-FLD method.
>>Reply: The information about STXs used have been added in sentence (P11 L348-350).
Reviewer 2 Report
General comments
The article describes the variation in saxitoxins (STXs) and tetrodotoxin (TTX) levels and profiles in xanthid crabs Zosimus aeneus collected in a single reef on Ishigaki Island in Japan. The samples were collected in 2018 and 2019 and the GPS coordinates were precisely recorded. Based on their results, the authors reported the existence of three distinct zones (northwestern, central and southeastern zones, respectively) in the reef characterized by differences in toxin profiles of the crab specimens. The analysis of the stomach content of samples collected in 2019 revealed the presence of ascidian spicules in large numbers and this led the authors to hypothesize that the toxic characteristics of the xanthid crabs may have been transferred from ascidians.
The hypothesis of ascidians being responsible for the toxication of the xanthid crabs is interesting and worth investigating further but for the time being, the authors should avoid stating with “confidence” that a link exists, as they did not prove it.
Experimental details are missing in the Materials and methods section.
Specific comments
- Abstract
- Line 17: the verb “possess” is not appropriate here. Replace with “contain” or any other suitable term. This has to be changed in several instances throughout the manuscript (Lines 47, 57, 230, 281)
- Lines 21-22: there is no verb in the sentence starting with “The STXs/TTX content…”
- Introduction
- Please add a figure with the structure of TTX and STXs, showing notably the main STX analogues and those found in the crab samples
- Line 70: Figure 1 is slightly blurred, especially the writings and numbers. The quality should be improved
- Results
- Lines 95-96: “… No. 9, which contained a relatively high TTX content”. Please indicate the percentage of STX in this sample, for comparison purposes with what is stated at the beginning of the sentence. This could be added in brackets at the end of the sentence (eg. X % STXs)
- Line 96: replace “…are comprise of…” with “include”
- Line 100: Please indicate the range of the ratio of neoSTX+hyneoSTX and STX+hySTX+dcSTX, in brackets, after “… varied among individuals…”
- Line 101: it is not clear whether the statement “total toxin” refers to the STXs (STX group + GTX group) or to the STXs+TTX
- Lines 104-105: remove the word “level” in the sentence
- Lines 110-111: in the sentence starting with “STXs comprising mainly…” the term “majority” is too vague. Please either indicate up to what percentage the STX group accounted for or the range of percentages
- Line 149: Please indicate the percentage of STXs for the samples No. 40 and 41, as they also contain TTX
- Lines 208-209: please remove the decimal places (92% and 70%) to be consistent with the way the results are presented in the previous sections of the manuscript
- Line 209-210: please indicate the percentages of As-2 and As-3
- Line 222: in figure 5, there is no need to add the type of spicule as labels, but if so please add As-4 as a label in the CTR graph. I would rather replace the spicule type as labels in the graphs with the corresponding percentages
- Discussion
- Line 227: replace “contains” with “contain”
- Lines 234-235: the authors mentioned that Lissoclinum are considered a good candidate source of STXs; yet, As-4 was also found in crabs from the CTR zone, although no GTXs were detected in the stomach content of aeneus in CTR (only TTX). This should be discussed in the manuscript
- Lines 251-255: It should be clarified here that viscera include stomach, heart, gonads, and intestine. This could also be a reason why the toxin characteristics between appendages and viscera (2018 data) are different from appendages and stomach (2019 data)
- Lines 273-275: Please discuss about the impact this transformation might have on the xanthid crabs, as STX group toxins are more toxic than GTXs. Are they affected by the toxin transformation?
- Line 280: I did not get the figure S1 along with the manuscript. How were the other TTX analogues identified, as the authors reported that they did not have the corresponding standards?
- Line 290: replace “content” with “presence”
- Line 297-299: There seems to be a missing part in the sentence starting with “However”
- Line 302: I do not agree with the term “confidence”. The authors should moderate their assertion here. This is still hypothetical and needs to be verified
- Materials and methods
- There is no information regarding the toxin standards used for quantification or simply for identification purposes
- Lines 321-322: The phrasing suggests that the crabs were still alive when the stomach was removed. Is it the case? If so, this was not considered as an ethical issue?
- Line 332: please indicate what quantity of tissue and what volume of HCl solution were used for the extraction
- Line 338: please indicate which STXs were searched for
- Others
- Lines 382-383: the supplementary material corresponding to figure S1 was not provided
Author Response
To Reviewer 2: 
Many thanks for your valuable comments. We revised our manuscript according to the comments, as indicated below (revised parts are indicated in blue font).
The article describes the variation in saxitoxins (STXs) and tetrodotoxin (TTX) levels and profiles in xanthid crabs Zosimus aeneus collected in a single reef on Ishigaki Island in Japan. The samples were collected in 2018 and 2019 and the GPS coordinates were precisely recorded. Based on their results, the authors reported the existence of three distinct zones (northwestern, central and southeastern zones, respectively) in the reef characterized by differences in toxin profiles of the crab specimens. The analysis of the stomach content of samples collected in 2019 revealed the presence of ascidian spicules in large numbers and this led the authors to hypothesize that the toxic characteristics of the xanthid crabs may have been transferred from ascidians.
The hypothesis of ascidians being responsible for the toxication of the xanthid crabs is interesting and worth investigating further but for the time being, the authors should avoid stating with “confidence” that a link exists, as they did not prove it.
Experimental details are missing in the Materials and methods section.
- Abstract
- Line 17: the verb “possess” is not appropriate here. Replace with “contain” or any other suitable term. This has to be changed in several instances throughout the manuscript (Lines 47, 57, 230, 281)
>>Reply: We have revised all parts you mentioned. Please see P1 L17, 52, 62, 227 and 281.
- Lines 21-22: there is no verb in the sentence starting with “The STXs/TTX content…”
>>Reply: We have added a verb to this sentence (P1 L22).
- Introduction
- Please add a figure with the structure of TTX and STXs, showing notably the main STX analogues and those found in the crab samples
>>Reply: Thank you for your suggestion. We have added the structure of TTX and STXs as Figure 1 in the manuscript. Please see Figure.1.
- Line 70: Figure 1 is slightly blurred, especially the writings and numbers. The quality should be improved
>>Reply: We have been changed the quality of the figure. Please see Figure 2.
- Results
- Lines 95-96: “… No. 9, which contained a relatively high TTX content”. Please indicate the percentage of STX in this sample, for comparison purposes with what is stated at the beginning of the sentence. This could be added in brackets at the end of the sentence (eg. X % STXs)
>>Reply: The percentage of STXs was provided in the sentence (P4 L100).
- Line 96: replace “…are comprise of…” with “include”
>>Reply: As you pointed out, we have changed "are comprise of" to "include"(P4 L100)..
- Line 100: Please indicate the range of the ratio of neoSTX+hyneoSTX and STX+hySTX+dcSTX, in brackets, after “… varied among individuals…”
>>Reply: We have been revised this part. Please see P4 L103-104.
- Line 101: it is not clear whether the statement “total toxin” refers to the STXs (STX group + GTX group) or to the STXs+TTX
>>Reply: We are sorry for the confusion. "total toxin" refers to STXs+TTXs. We have corrected this section and added (STXs+TTX) after ”total toxin” (P4 L105).
- Lines 104-105: remove the word “level” in the sentence
>>Reply: We have removed the "level" in the sentence as you suggested (P4 L108).
- Lines 110-111: in the sentence starting with “STXs comprising mainly…” the term “majority” is too vague. Please either indicate up to what percentage the STX group accounted for or the range of percentages
>>Reply: We have indicated in the sentence what percentage the STX group specifically accounted for. (P4 L114-115).
- Line 149: Please indicate the percentage of STXs for the samples No. 40 and 41, as they also contain TTX
>>Reply: We have indicated the percentage of STX in No.40 and No.41 (P6 L153-154).
- Lines 208-209: please remove the decimal places (92% and 70%) to be consistent with the way the results are presented in the previous sections of the manuscript
>>Reply: We have removed the decimal places (P8 L213-214).
- Line 209-210: please indicate the percentages of As-2 and As-3
>>Reply: We have indicated the percentages of As-2 and As-3 (P8 L213-214).
- Line 222: in figure 5, there is no need to add the type of spicule as labels, but if so please add As-4 as a label in the CTR graph. I would rather replace the spicule type as labels in the graphs with the corresponding percentages
>>Reply: We have revised the figure. Please see Figure 6.
- Discussion
- Line 227: replace “contains” with “contain”
>>Reply: We have replaced "contains" with "contain" (P9 L222).
- Lines 234-235: the authors mentioned that Lissoclinum are considered a good candidate source of STXs; yet, As-4 was also found in crabs from the CTR zone, although no GTXs were detected in the stomach content of aeneus in CTR (only TTX). This should be discussed in the manuscript
>>Reply: Thank you for your comments. We have been revised the discussion of this part in the manuscript. Please see L297-302. We hope you can agree with our revision.
- Lines 251-255: It should be clarified here that viscera include stomach, heart, gonads, and intestine. This could also be a reason why the toxin characteristics between appendages and viscera (2018 data) are different from appendages and stomach (2019 data)
>>Reply: Thank you for your valuable suggestion. We have edited this part. please see P9 L254-255.
- Lines 273-275: Please discuss about the impact this transformation might have on the xanthid crabs, as STX group toxins are more toxic than GTXs. Are they affected by the toxin transformation?
>>Reply: Toxin conversion has been confirmed in this study and previous studies, but the effect of this component conversion on crabs is unknown. However, as you mentioned, we have been revised this part of the discussion in the manuscript (P10 L272-275).
- Line 280: I did not get the figure S1 along with the manuscript. How were the other TTX analogues identified, as the authors reported that they did not have the corresponding standards?
>>Reply: We have sent the supplementary materials to the Marine Drugs Editorial Office, when we submitted the manuscript. We used MRM transitions and compared the retention time with the TTX and previous papers. We have been added the m/z information for some TTX analogues in the sentence (P10 L280-281).
- Line 290: replace “content” with “presence”
>>Reply: We have replaced “content” with “presence”(P10 L291).
- Line 297-299: There seems to be a missing part in the sentence starting with “However”
>>Reply: We have revised the sentence (P10 L298-303)
- Line 302: I do not agree with the term “confidence”. The authors should moderate their assertion here. This is still hypothetical and needs to be verified
>>Reply: We have changed the wording a little in response to your suggestion (P10 L303).
- Materials and methods
- There is no information regarding the toxin standards used for quantification or simply for identification purposes
>>Reply: We have added the information about the standards (P11 L348-350 and P12 L367-368)
- Lines 321-322: The phrasing suggests that the crabs were still alive when the stomach was removed. Is it the case? If so, this was not considered as an ethical issue?
>>Reply: Thank you for your comments. The crabs were killed while paralyzed in ice water before being used in the experiment. In Japan, there are no specific regulations for animal testing on crustaceans, but we are very careful about the ethical aspects of using them in our experiments. We hope that you will agree with our explanation.
- Line 332: please indicate what quantity of tissue and what volume of HCl solution were used for the extraction
>>Reply: The sentence has modified to indicate the relationship between the quantity of tissue and the volume of HCl were used for extraction (P11 L331)
Line 338: please indicate which STXs were searched for
>>Reply: We have added the information about STXs used (P11 L348-350).
- Others
- Lines 382-383: the supplementary material corresponding to figure S1 was not provided
>>Reply: We have sent the supplementary materials to the Marine Drugs Editorial Office when submitting the manuscript.
Round 2
Reviewer 1 Report
Thanks for your responses.
Author Response
Thank you very much for your valuable comments. As you pointed out, we understand that it is important to show the chromatogram of STX (PST), which is the main component of crab toxin. Therefore, we have added the chromatogram of PST toxin as Figure 7 in "Discussion" as a typical analysis.
We are always trying to use LC-MS/MS for the analysis of STXs. However, we have not performed LC-MS/MS analysis for STXs in this study because the conditions for clear separation of each component of STXs using a HILIC column have not been established and the detection sensitivity of MS is inferior to that of HPLC-FLD.
Assuming that the ionization efficiency per unit amount of the other analogues was the same as that of TTX, the relative proportions of TTX-related substances were determined and explanations were added to Figure S1.